# Boost then Convolve: Gradient Boosting Meets Graph Neural Networks

**Sergei Ivanov**
Criteo AI Lab; Skoltech
Paris, France
s.ivanov@criteo.com

**Liudmila Prokhorenkova**
Yandex; HSE University; MIPT
Moscow, Russia
ostroumova-la@yandex-team.ru

## Abstract

Graph neural networks (GNNs) are powerful models that have been successful in various graph representation learning tasks. Whereas gradient boosted decision trees (GBDT) often outperform other machine learning methods when faced with heterogeneous tabular data. But what approach should be used for graphs with tabular node features? Previous GNN models have mostly focused on networks with homogeneous sparse features and, as we show, are suboptimal in the heterogeneous setting. In this work, we propose a novel architecture that trains GBDT and GNN jointly to get the best of both worlds: the GBDT model deals with heterogeneous features, while GNN accounts for the graph structure. Our model benefits from end-to-end optimization by allowing new trees to fit the gradient updates of GNN. With an extensive experimental comparison to the leading GBDT and GNN models, we demonstrate a significant increase in performance on a variety of graphs with tabular features. The code is available: https://github.com/nd7141/bgnn.

## 1 Introduction

Graph neural networks (GNNs) have shown great success in learning on graph-structured data with various applications in molecular design (Stokes et al., 2020), computer vision (Casas et al., 2019), combinatorial optimization (Mazyavkina et al., 2020), and recommender systems (Sun et al., 2020). The main driving force for progress is the existence of canonical GNN architecture that efficiently encodes the original input data into expressive representations, thereby achieving high-quality results on new datasets and tasks.

Recent research has mostly focused on GNNs with sparse data representing either homogeneous node embeddings (e.g., one-hot encoded graph statistics) or bag-of-words representations. Yet tabular data with detailed information and rich semantics among nodes in the graph are more natural for many situations and abundant in real-world AI (Xiao et al., 2019). For example, in a social network, each person has socio-demographic characteristics (e.g., age, gender, date of graduation), which largely vary in data type, scale, and missing values. GNNs for graphs with tabular data remain unexplored, with gradient boosted decision trees (GBDTs) largely dominating in applications with such heterogeneous data (Bentéjac et al., 2020).

GBDTs are so successful for tabular data because they possess certain properties: (i) they efficiently learn decision space with hyperplane-like boundaries that are common in tabular data; (ii) they are well-suited for working with variables of high cardinality, features with missing values, and of different scale; (iii) they provide qualitative interpretation for decision trees (e.g., by computing decrease in node impurity for every feature) or for ensembles via post-hoc analysis stage (Kaur et al., 2020); (iv) in practical applications, they mostly converge faster even for large amounts of data.

In contrast, a crucial feature of GNNs is that they take into account both the neighborhood information of the nodes and the node features to make a prediction, unlike GBDTs that require additional preprocessing analysis to provide the algorithm with graph summary (e.g., through unsupervised graph embeddings (Hu et al., 2020a)). Moreover, it has been shown theoretically that message-passing GNNs can compute any function on its graph input that is computable by a Turing machine, i.e., GNN is known to be the only learning architecture that possesses universality properties on graphs (approximation (Keriven & Peyré, 2019; Maron et al., 2019) and computability (Loukas, 2020)).

Furthermore, gradient-based learning of neural networks can have numerous advantages over the tree-based approach: (i) relational inductive bias imposed in GNNs alleviates the need to manually engineer features that capture the topology of the network (Battaglia et al., 2018); (ii) the end-to-end nature of training neural networks allows multi-stage (Fey et al., 2019) or multi-component (Wang et al., 2020) integration of GNNs in application-dependent solutions; (iii) pretraining representations with graph networks enriches transfer learning for many valuable tasks such as unsupervised domain adaptation (Wu et al., 2020), self-supervised learning (Hu et al., 2020b), and active learning regimes (Satorras & Estrach, 2018).

Undoubtedly, there are major benefits in both GBDT and GNN methods. Is it possible to get advantages of both worlds? All previous approaches (Arik & Pfister, 2020; Popov et al., 2019; Badirli et al., 2020) that attempt to combine gradient boosting and neural networks are computationally heavy, do not consider graph-structured data, and suffer from the lack of relational bias imposed in GNN architectures, see Appendix A for a more detailed comparison with related literature. To the best of our knowledge, the current work is the first to explore using GBDT models for graph-structured data.

In this paper, we propose a novel learning architecture for graphs with tabular data, **BGNN**, that combines GBDT's learning on tabular node features with GNN that refines the predictions utilizing the graph's topology. This allows **BGNN** to inherit the advantages of gradient boosting methods (heterogeneous learning and interpretability) and graph networks (representation learning and end-to-end training). Overall, our contributions are the following:

(1) We design a novel generic architecture that combines GBDT and GNN into a unique pipeline. To the best of our knowledge, this is the first work that systematically studies the application of GBDT to graph-structured data.

(2) We overcome the challenge of end-to-end training of GBDT by iteratively adding new trees that fit the gradient updates of GNN. This allows us to backpropagate the error signal from the topology of the network to GBDT.

(3) We perform an extensive evaluation of our approach against strong baselines in node prediction tasks. Our results consistently demonstrate significant performance improvements on heterogeneous node regression and node classification tasks over a variety of real-world graphs with tabular data.

(4) We show that our approach is also more efficient than the state-of-the-art GNN models due to much faster loss convergence during training. Furthermore, learned representations exhibit discernible structure in the latent space, which further demonstrates the expressivity of our approach.

## 2 BACKGROUND

Let $G = (V, E)$ be a graph with nodes having features and target labels. In node prediction tasks (classification or regression), some target labels are known, and the goal is to predict the remaining ones. Throughout the text, by lowercase variables $\mathbf{x}_v$ ($v \in V$) or $\mathbf{x}$ we denote features of individual nodes, and $\mathbf{X}$ represents the matrix of all features for $v \in V$. Individual target labels are denoted by $y_v$, while $Y$ is the vector of known labels.

**Graph Neural Networks (GNNs)** use both the network's connectivity and the node features to learn latent representations for all nodes $v \in V$. Many popular GNNs use a neighborhood aggregation approach, also called the message-passing mechanism, where the representation of a node $v$ is updated by applying a non-linear aggregation function of $v$'s neighbors representation (Fey & Lenssen, 2019). Formally, GNN is a differentiable, permutation-invariant function $g_\theta : (G, \mathbf{X}) \mapsto \widehat{Y}$, where $\widehat{Y}$ is the vector of predicted labels. Similar to traditional neural networks, GNNs are composed of multiple layers, each representing a non-linear message-passing function:

$$\mathbf{x}_v^t = \text{COMBINE}^t \left( \mathbf{x}_v^{t-1}, \text{AGGREGATE}^t \left( \left\{ (\mathbf{x}_w^{t-1}, \mathbf{x}_v^{t-1}) : (w, v) \in E \right\} \right) \right), \quad (1)$$

where $\mathbf{x}_v^t$ is the representation of node $v$ at layer $t$, and COMBINE$^t$ and AGGREGATE$^t$ are (parametric) functions that aggregate representations from the local neighborhood of a node. Then, the GNN mapping $g_\theta$ includes multiple layers of aggregation (1). Parameters of GNN model $\theta$ are optimized with gradient descent by minimizing an empirical loss function $L_{\text{GNN}}(Y, g_\theta(G, \mathbf{X}))$.

**Gradient Boosted Decision Trees (GBDT)** is a well-known and widely used algorithm that is defined on non-graph tabular data (Friedman, 2001) and is particularly successful for tasks containing heterogeneous features and noisy data.

The core idea of gradient boosting is to construct a strong model by iteratively adding weak ones (usually decision trees). Formally, at each iteration $t$ of the gradient boosting algorithm, the model $f(\mathbf{x})$ is updated in an additive manner:

$$f^t(\mathbf{x}) = f^{t-1}(\mathbf{x}) + \epsilon\, h^t(\mathbf{x}), \tag{2}$$

where $f^{t-1}$ is a model constructed at the previous iteration, $h^t$ is a weak learner that is chosen from some family of functions $\mathcal{H}$, and $\epsilon$ is a learning rate. The weak learner $h^t \in \mathcal{H}$ is chosen to approximate the negative gradient of a loss function $L$ w.r.t. the current model's predictions:

$$h^t = \underset{h \in \mathcal{H}}{\arg\min} \sum_i \left( -\frac{\partial L(f^{t-1}(\mathbf{x}_i), y_i)}{\partial f^{t-1}(\mathbf{x}_i)} - h(\mathbf{x}_i) \right)^2. \tag{3}$$

The gradient w.r.t. the current predictions indicates how one should change these predictions to improve the loss function. Informally, gradient boosting can be thought of as performing gradient descent in functional space.

The set of weak learners $\mathcal{H}$ is usually formed by shallow decision trees. Decision trees are built by a recursive partition of the feature space into disjoint regions called leaves. This partition is usually constructed greedily to minimize the loss function (3). Each leaf $R_j$ of the tree is assigned to a value $a_j$, which estimates the response $y$ in the corresponding region. In our case, $a_j$ is equal to the average negative gradient value in the leaf $R_j$. To sum up, we can write $h(x) = \sum_j a_j 1_{\{x \in R_j\}}$.

## 3 GBDT MEETS GNN

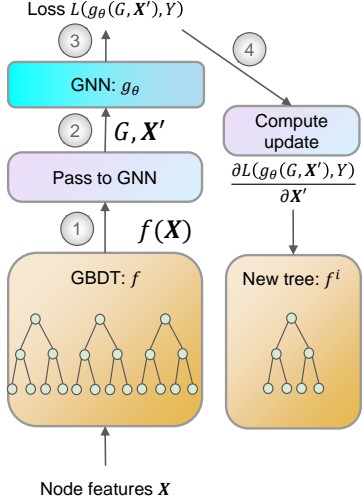

Figure 1: Training of **BGNN**, steps for one epoch are numbered.

---

**Algorithm 1** Training of **BGNN**

**Input:** Graph $G$, node features $\mathbf{X}$, targets $Y$
Initialize GBDT targets $\mathcal{Y} = Y$
**for** epoch $i = 1$ to $N$ **do**
    # Train $k$ trees of GBDT with eq. (2)-(3)
    $f^i \leftarrow \underset{k}{} \underset{f^i}{\arg\min} L_{\text{GBDT}}(f^i(\mathbf{X}), \mathcal{Y})$
    $f \leftarrow f + f^i$

    # Train $l$ steps of GNN on new node features
    $\mathbf{X}' \leftarrow f(\mathbf{X})$
    $\theta, \mathbf{X}' \leftarrow \underset{l}{} \underset{\theta, \mathbf{X}'}{\arg\min} L_{\text{GNN}}(g_\theta(G, \mathbf{X}'), Y)$

    # Update targets for next iteration of GBDT
    $\mathcal{Y} \leftarrow \mathbf{X}' - f(\mathbf{X})$
**end for**
**Output:** Models GBDT $f$ and GNN $g_\theta$

---

Gradient boosting approach is successful for learning on tabular data; however, there are challenges of applying GBDT on graph-structured data: (i) how to propagate relational signal, in addition to node features, to otherwise inherently tabular model; and (ii) how to train it together with GNN in an end-to-end fashion. Indeed, optimizations of GBDT and GNN follow different approaches: the parameters of GNN are optimized via gradient descent, while GBDT is constructed iteratively, and the decision trees remain fixed after being built (decision trees are based on hard splits of the feature space, which makes them non-differentiable).

A straightforward approach would be to train the GBDT model only on the node features and then use the obtained predictions, jointly with the original input, as new node features for GNN. In this case, the graph-insensitive predictions of GBDT will further be refined by a graph neural network.

This approach (which we call **Res-GNN**) can already boost the performance of GNN for some tasks. However, in this case, the GBDT model completely ignores the graph structure and may miss descriptive features of the graph, providing inaccurate input data to GNN.

In contrast, we propose end-to-end training of GBDT and GNN called **BGNN** (for Boost-GNN). As before, we first apply GBDT and then GNN, but now we optimize both of them, taking into account the quality of final predictions. The training of **BGNN** is shown in Figure 1. Recall that one cannot tune already built decision trees due to their discrete structure, so we iteratively update the GBDT model by adding new trees that approximate the GNN loss function.

In Algorithm 1, we present the training of **BGNN** that combines GBDT and GNN for any node-level prediction problem such as semi-supervised node regression or classification. In the first iteration, we build a GBDT model $f^1(\mathbf{x})$ with $k$ decision trees by minimizing the loss function $L_{\mathrm{GBDT}}(f^1(\mathbf{x}), y)$ (e.g., RMSE for regression or cross-entropy for classification) averaged over the train nodes, following the equations (2)-(3). Using all predictions $f^1(\mathbf{X})$, we update the node features to $\mathbf{X}'$ that we pass to GNN. Possible update functions that we experiment with include concatenation with the original node features and their replacement by $f^1(\mathbf{X})$. Next, we train a graph neural network $g_\theta$ on a graph $G$ with node features $\mathbf{X}'$ by minimizing $L_{\mathrm{GNN}}(g_\theta(G, \mathbf{X}'), Y)$ with $l$ steps of gradient descent.[1] Importantly, we optimize both the parameters $\theta$ of GNN and the node features $\mathbf{X}'$. Then, we use the difference between the optimized node features $\mathbf{X}'_{new}$ and the input node features $\mathbf{X}' = f^1(\mathbf{X})$ as the target for the next decision trees built by GBDT. If $l = 1$, the difference $\mathbf{X}'_{new} - \mathbf{X}'$ exactly equals the negative gradient of the loss function w.r.t. the input features $\mathbf{X}'$ multiplied by the learning rate $\eta$:

$$\mathbf{X}'_{new} = \mathbf{X}' - \eta \frac{\partial L_{\mathrm{GNN}}(g_\theta(G, \mathbf{X}'), Y)}{\partial \mathbf{X}'} \,.$$

In the second iteration, we train a new GBDT model $f^2$ with the original input features $\mathbf{X}$ but new target labels: $\mathbf{X}'_{new} - \mathbf{X}'$. Intuitively, $f^2$ fits the direction that would improve GNN prediction based on the first predictions $f^1(\mathbf{X})$. In other words, GBDT approximates the gradient steps made by GNN for the node features $\mathbf{X}'$. This is a regression problem, so here $L_{\mathrm{GBDT}}$ is the RMSE loss.

After $f^2$ is trained, we combine the predictions $f(\mathbf{X}) = f^1(\mathbf{X}) + f^2(\mathbf{X})$ and pass the obtained values $\mathbf{X}'$ to GNN as node features. GNN model $g_\theta$ again does $l$ steps of backpropagation and passes the new difference $\mathbf{X}'_{new} - \mathbf{X}'$ as a target to the next iteration of GBDT. In total, the model is trained for $N$ epochs and outputs a GBDT model $f : \mathbf{X} \mapsto Y$ and GNN model $g_\theta : (G, \mathbf{X}) \mapsto Y$, which can be used for downstream tasks.

Intuitively, **BGNN** model consists of two consecutive blocks, GBDT and GNN, which are trained end-to-end, and therefore can be interpreted from two angles: GBDT as an embedding layer for GNN or GNN as a parametric loss function for GBDT. In the former case, GBDT transforms the original input features $\mathbf{X}$ to new node features $\mathbf{X}'$, which are then passed to GNN. In the latter case, one can see **BGNN** as a standard gradient boosted training where GNN acts as a complex loss function that depends on the graph topology.

## 4 EXPERIMENTS

We have performed a comparative evaluation of **BGNN** and **Res-GNN** against a wide variety of strong baselines and previous approaches on heterogeneous node prediction problems, achieving significant improvement in performance across all of them. This section outlines our experimental setting, the results on node regression and classification problems, and extracted feature representations.

In our first experiments, we want to answer two questions:

Q1 *Does combination of GBDT and GNN lead to better qualitative results in heterogeneous node regression and classification problems?*

Q2 *Is the end-to-end training proposed in Algorithm 1 better than a combination of pretrained GBDT with GNN?*

To answer these questions, we consider several strong baselines among GBDTs, GNNs, and pure neural networks. **CatBoost** is a recent GBDT implementation (Prokhorenkova et al., 2018) that uses

---

[1] $L_{\mathrm{GNN}}$ is determined by the final task, e.g., RMSE for regression or the cross-entropy loss for classification.

oblivious trees as weak learners. **LightGBM** is another GBDT model (Ke et al., 2017) that is used extensively in ML competitions. Among GNNs, we tested four state-of-the-art recent models that showed superior performance in node prediction tasks: **GAT** (Veličković et al., 2018), **GCN** (Kipf & Welling, 2017), **AGNN** (Thekumparampil et al., 2018), **APPNP** (Klicpera et al., 2019). Additionally, we test the performance of fully-connected neural network **FCNN** and its end-to-end combination with GNNs, **FCNN-GNN**.

We compare these baselines against two proposed approaches: the end-to-end **BGNN** model and not end-to-end **Res-GNN**. The **BGNN** model follows Algorithm 1 and builds each tree approximating the **GNN** error in the previous iteration. In contrast, **Res-GNN** first trains a GBDT model on the training set of nodes and then either appends its predictions for all nodes to the original node features or replaces the original features with the GBDT predictions, after which GNN is trained on the updated features, and GNN's predictions are used to calculate metrics. Hence, **Res-GNN** is a two-stage approach where the training of GBDT is independent of GNN. On the other hand, **BGNN** trains GBDT and GNN simultaneously in an end-to-end fashion. In most of our experiments, the GNN-component of **FCNN-GNN**, **Res-GNN**, and **BGNN** is based on **GAT**, while in Section 4.3 we analyze consistency of improvements across different GNN models.

We ensure that the comparison is done fairly by training each model until the convergence with a reasonable set of hyperparameters evaluated on the validation set. We run each hyperparameter setting three times and take the average of the results. Furthermore, we have five random splits of the data, and the final number represents the average performance of the model for all five random seeds. More details about hyperparameters can be found in Appendix B.

## 4.1 NODE REGRESSION

### 4.1.1 DATASETS

We utilize five real-world node regression datasets with different properties outlined in Table 1. Four of these datasets are heterogeneous, i.e., the input features are of different types, scales, and meaning. For example, for the **VK** dataset, the node features are both numerical (e.g., last time seen on the platform) and categorical (e.g., country of living and university). On the other hand, **Wiki** dataset is homogeneous, i.e., the node features are interdependent and correspond to the bag-of-words representations of Wikipedia articles. Additional details about the datasets can be found in Appendix C.

Table 1: Summary of regression datasets.

|  | House | County | VK | Avazu | Wiki |
|---|---|---|---|---|---|
| **Setting** | Heterogeneous | Heterogeneous | Heterogeneous | Heterogeneous | Homogeneous |
| **# Nodes** | 20640 | 3217 | 54028 | 1297 | 5201 |
| **# Edges** | 182146 | 12684 | 213644 | 54364 | 198493 |
| **# Features/Node** | 6 | 7 | 14 | 9 | 3148 |
| **Mean Target** | 2.06 | 5.44 | 35.47 | 0.08 | 27923.86 |
| **Min Target** | 0.14 | 1.7 | 13.48 | 0 | 16 |
| **Max Target** | 5.00 | 24.1 | 118.39 | 1 | 849131 |
| **Median Target** | 1.79 | 5 | 33.83 | 0 | 9225 |

### 4.1.2 RESULTS

The results of our comparative evaluation for node regression are summarized in Table 2. We report the mean RMSE (with standard deviation) on the test set and the relative gap between RMSE of the **GAT** model (Veličković et al., 2018) and other methods, i.e., gap $= (r_m - r_{gnn})/r_{gnn}$, where $r_m$ and $r_{gnn}$ are RMSE of that model and of **GAT**, respectively.

Our results demonstrate significant improvement of **BGNN** over the baselines. In particular, in the heterogeneous case, **BGNN** achieves 8%, 14%, 4%, and 4% reduction of the error for **House**, **County**, **VK**, and **Avazu** datasets, respectively. **Res-GNN** model that uses a pretrained **CatBoost** model for the input of **GNN** also decreases RMSE, although not as much as the end-to-end model

Table 2: Summary of our results for node regression. Gap % is relative difference w.r.t. **GAT** RMSE (the smaller the better). Top-2 results are highlighted in **bold**.

| | Method | Heterogeneous | | | | | | | | | | Homogeneous | |
| | | House | | County | | VK | | Avazu | | | | Wiki | |
| | | RMSE | Gap % | RMSE | Gap % | RMSE | Gap % | RMSE | Gap % | | | RMSE | Gap % |
|---|---|---|---|---|---|---|---|---|---|---|---|---|---|
| GBDT | CatBoost | $0.63 \pm 0.01$ | 15.3 | $1.39 \pm 0.07$ | -4.32 | $7.16 \pm 0.20$ | -0.82 | $0.1172 \pm 0.02$ | 3.36 | | | $46359 \pm 4508$ | 0.97 |
| | LightGBM | $0.63 \pm 0.01$ | 15.98 | $1.4 \pm 0.07$ | -3.93 | $7.2 \pm 0.21$ | -0.33 | $0.1171 \pm 0.02$ | 3.27 | | | $49915 \pm 3643$ | 8.71 |
| GNN | GAT | $0.54 \pm 0.01$ | 0 | $1.45 \pm 0.06$ | 0 | $7.22 \pm 0.19$ | 0 | $0.1134 \pm 0.01$ | 0 | | | $\mathbf{45916 \pm 4527}$ | **0** |
| | GCN | $0.63 \pm 0.01$ | 16.77 | $1.48 \pm 0.08$ | 2.06 | $7.25 \pm 0.19$ | 0.34 | $0.1141 \pm 0.02$ | 0.58 | | | $\mathbf{44936 \pm 4083}$ | **-2.14** |
| | AGNN | $0.59 \pm 0.01$ | 8.01 | $1.45 \pm 0.08$ | -0.19 | $7.26 \pm 0.20$ | 0.54 | $0.1134 \pm 0.02$ | -0.02 | | | $45982 \pm 3058$ | 0.14 |
| | APPNP | $0.69 \pm 0.01$ | 27.11 | $1.5 \pm 0.11$ | 3.39 | $13.23 \pm 0.12$ | 83.19 | $0.1127 \pm 0.01$ | -0.65 | | | $53426 \pm 4159$ | 16.36 |
| NN | FCNN | $0.68 \pm 0.02$ | 25.49 | $1.48 \pm 0.07$ | 1.56 | $7.29 \pm 0.21$ | 1.02 | $0.118 \pm 0.02$ | 4.07 | | | $51662 \pm 2983$ | 12.51 |
| | FCNN-GNN | $0.53 \pm 0.01$ | -2.48 | $1.39 \pm 0.06$ | -4.68 | $7.22 \pm 0.20$ | 0.01 | $0.1114 \pm 0.02$ | -1.82 | | | $48491 \pm 7889$ | 5.61 |
| Ours | Res-GNN | $\mathbf{0.51 \pm 0.01}$ | **-6.39** | $\mathbf{1.33 \pm 0.08}$ | **-8.35** | $\mathbf{7.07 \pm 0.20}$ | **-2.04** | $\mathbf{0.1095 \pm 0.01}$ | **-3.42** | | | $46747 \pm 4639$ | 1.81 |
| | BGNN | $\mathbf{0.5 \pm 0.01}$ | **-8.15** | $\mathbf{1.26 \pm 0.08}$ | **-13.67** | $\mathbf{6.95 \pm 0.21}$ | **-3.8** | $\mathbf{0.109 \pm 0.01}$ | **-3.9** | | | $49222 \pm 3743$ | 7.2 |

**BGNN**. In the homogeneous dataset **Wiki**, **CatBoost** and, subsequently, **Res-GNN** and **BGNN** are outperformed by the **GNN** model. Intuitively, when the features are homogeneous, neural network approaches are sufficient to attain the best results. This shows that **BGNN** *leads to better qualitative results and its end-to-end training outperforms other approaches in node prediction tasks for graphs with heterogeneous tabular data*.

We can also observe that the end-to-end combination **FCNN**-**GNN** often leads to better performance than pure **GNN**. However, its improvement is smaller than for **BGNN** which uses the advantages of GBDT models. Moreover, **CatBoost** and **LightGBM** can be effective on their own, but their performance is not stable across all datasets. Overall, these experiments demonstrate the superiority of **BGNN** against other strong models.

## 4.2 NODE CLASSIFICATION

For node classification, we use five datasets with different properties. Due to the lack of publicly available datasets with heterogeneous node features, we adopt the datasets **House_class** and **VK_class** from the regression task by converting the target labels into several discrete classes. We additionally include two sparse node classification datasets **SLAP** and **DBLP** coming from heterogeneous information networks (HIN) with nodes having different types. We also include one homogeneous dataset **OGB-ArXiv** (Hu et al., 2020a). In this dataset, the node features correspond to a 128-dimensional feature vector obtained by averaging the embeddings of words in the title and abstract. Hence, the features are not heterogeneous, and therefore GBDT is not expected to outperform neural network approaches. More details about these datasets can be found in Appendix D.

Table 3: Summary of our results for node classification. Gap % is the relative difference w.r.t. **GAT** accuracy (the higher the better). Top-2 results are highlighted in **bold**.

| | Method | Heterogeneous | | | | | | | | Homogeneous | |
| | | House_class | | VK_class | | Slap | | DBLP | | OGB-ArXiv | |
| | | Acc. | Gap % | Acc. | Gap % | Acc. | Gap % | Acc. | Gap % | Acc. | Gap % |
|---|---|---|---|---|---|---|---|---|---|---|---|
| GBDT | CatBoost | $0.52 \pm 0.01$ | -16.82 | $0.57 \pm 0.01$ | -1.26 | $0.922 \pm 0.01$ | 15.12 | $0.759 \pm 0.03$ | -5.42 | 0.45 | -36.35 |
| | LightGBM | $0.55 \pm 0.00$ | -11.98 | $0.579 \pm 0.01$ | 0.26 | $\mathbf{0.963 \pm 0.00}$ | **20.3** | $\mathbf{0.913 \pm 0.01}$ | **13.73** | 0.51 | -26.97 |
| GNN | GAT | $0.625 \pm 0.00$ | 0 | $0.577 \pm 0.00$ | 0 | $0.801 \pm 0.01$ | 0 | $0.802 \pm 0.01$ | 0 | **0.70** | **0** |
| | GCN | $0.6 \pm 0.00$ | -3.98 | $0.574 \pm 0.00$ | -0.6 | $0.878 \pm 0.01$ | 9.72 | $0.428 \pm 0.04$ | -46.6 | - | - |
| | AGNN | $0.614 \pm 0.01$ | -1.73 | $0.572 \pm 0.00$ | -0.79 | $0.892 \pm 0.01$ | 11.47 | $0.794 \pm 0.01$ | -1.02 | - | - |
| | APPNP | $0.619 \pm 0.00$ | -0.89 | $0.573 \pm 0.00$ | -0.67 | $0.895 \pm 0.01$ | 11.79 | $0.83 \pm 0.02$ | 3.47 | - | - |
| NN | FCNN | $0.534 \pm 0.01$ | -14.53 | $0.567 \pm 0.01$ | -1.72 | $0.759 \pm 0.04$ | -5.24 | $0.623 \pm 0.02$ | -22.3 | 0.50 | -28.91 |
| | FCNN-GNN | $\mathbf{0.64 \pm 0.00}$ | **2.36** | $0.589 \pm 0.00$ | 2.13 | $0.89 \pm 0.01$ | 11.11 | $0.81 \pm 0.01$ | 0.94 | **0.71** | **0.54** |
| Ours | Res-GNN | $0.625 \pm 0.01$ | -0.06 | $\mathbf{0.603 \pm 0.00}$ | **4.45** | $0.905 \pm 0.01$ | 13.06 | $\mathbf{0.892 \pm 0.01}$ | **11.11** | 0.70 | -0.33 |
| | BGNN | $\mathbf{0.682 \pm 0.00}$ | **9.18** | $\mathbf{0.683 \pm 0.00}$ | **18.3** | $\mathbf{0.95 \pm 0.00}$ | **18.61** | $0.889 \pm 0.01$ | 10.77 | 0.67 | -4.36 |

As can be seen from Table 3, on the datasets with heterogeneous tabular features (**House_class** and **VK_class**), **BGNN** outperforms other approaches with a significant margin. For example, for the

**VK_class** dataset **BGNN** achieves more than 18% of relative increase in accuracy. This demonstrates that learned representations of GBDT together with GNN can be equally useful for node classification setting on data with heterogeneous features.

The other two datasets, **Slap** and **DBLP**, have sparse bag-of-words features that are particularly challenging for the **GNN** model. On these two datasets, **GBDT** is the strongest baseline. Moreover, since **FCNN** outperforms **GNN**, we conclude that graph structure does not help, hence **BGNN** is not supposed to beat **GBDT**. This is indeed the case: the final accuracy of **BGNN** is slightly worse than that of **GBDT**.

In the homogeneous **OGB-ArXiv** dataset, **FCNN**-**GNN** and **GNN** achieve the top performance followed by **Res-GNN** and **BGNN** models.[2] In a nutshell, **GBDT** does not learn good predictions on the homogeneous input features and therefore reduces the discriminative power of **GNN**. Both cases, with sparse and with homogeneous features, show that the performance of **BGNN** is on par or higher than of **GNN**; however, lacking heterogeneous structure in the data may make the joint training of **GBDT** and **GNN** redundant.

## 4.3 CONSISTENCY ACROSS DIFFERENT GNN MODELS

Seeing that our models perform significantly better than strong baselines on various datasets, we want to test whether the improvement is consistent if different GNN models are used. Thus, we ask:

Q3 *Do different GNN models benefit from our approach of combination with GBDT?*

To answer this question, we compare GNN models that include **GAT** (Veličković et al., 2018), **GCN** (Kipf & Welling, 2017), **AGNN** (Thekumparampil et al., 2018), and **APPNP** (Klicpera et al., 2019). We substitute each of these models to **Res-GNN** and **BGNN** and measure the relative change in performance with respect to the original GNN's performance.

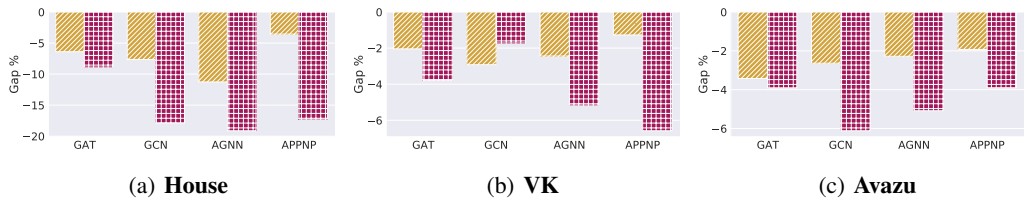

(a) **House**          (b) **VK**          (c) **Avazu**

Figure 2: Relative difference for **Res-GNN** (yellow, diagonal) and **BGNN** (red, squared) for different GNN architectures w.r.t. GNN RMSE (the smaller the better).

In Figure 2 we report the relative RMSE gap between **Res-GNN** and **BGNN** for each of the GNN architectures, i.e., we compute $gap = (r_m - r_{gnn})/r_{gnn}$, where $r_m$ and $r_{gnn}$ are RMSE of that model and of GNN respectively. This experiment positively answers Q3 and shows that *all tested GNN architectures significantly benefit from the proposed approach*. For example, for **House** dataset the decrease in the mean squared error is 9%, 18%, 19%, and 17% for **GAT**, **GCN**, **AGNN**, and **APPNP** models respectively. Additionally, one can see that the end-to-end training of **BGNN** (red, squared) leads to larger improvements than a naïve combination of **CatBoost** and **GNN** in **Res-GNN** (yellow, diagonal). Exact metrics and training time are in Appendix E.

## 4.4 TRAINING TIME

As the previous experiments demonstrated superior quality across various datasets and GNN models, it is important to understand if the additional GBDT part can become a bottleneck in terms of efficiency for training this model on real-world datasets. Hence, we ask:

Q4 *Do **BGNN** and **Res-GNN** models incur a significant increase in training time?*

---

[2]For **OGB-ArXiv** we used a different implementation of the GAT model to align results with the publicly available leaderboard: `https://ogb.stanford.edu/docs/nodeprop/`

To answer this question, we measure the clock time to train each model until convergence, considering early stopping. Table 4 presents training time for each model. We can see that both **BGNN** and **Res-GNN** run faster than **GNN** in most cases. In other words, **BGNN** and **Res-GNN** models *do not incur an increase in training time* but actually are more efficient than **GNN**. For example, for **VK** dataset **BGNN** and **Res-GNN** run 3x and 2x faster than **GNN**, respectively. Moreover, **BGNN** is consistently faster than another end-to-end implementation **FCNN-GNN** that uses **FCNN** instead of **CatBoost** to preprocess the original input features.

Table 4: Training time (s) in node regression task.

| | Method | House | County | VK | Wiki | Avazu |
|---|---|---|---|---|---|---|
| GBDT | **CatBoost** | $4 \pm 1$ | $2 \pm 1$ | $24 \pm 4$ | $10 \pm 1$ | $2 \pm 2$ |
| | **LightGBM** | $3 \pm 0$ | $1 \pm 0$ | $5 \pm 3$ | $3 \pm 2$ | $0 \pm 0$ |
| GNN | **GAT** | $35 \pm 2$ | $19 \pm 6$ | $42 \pm 4$ | $15 \pm 1$ | $9 \pm 2$ |
| | **GCN** | $28 \pm 0$ | $18 \pm 7$ | $38 \pm 0$ | $13 \pm 3$ | $12 \pm 6$ |
| | **AGNN** | $38 \pm 5$ | $28 \pm 3$ | $48 \pm 3$ | $19 \pm 5$ | $14 \pm 8$ |
| | **APPNP** | $68 \pm 1$ | $34 \pm 10$ | $81 \pm 3$ | $49 \pm 26$ | $24 \pm 15$ |
| NN | **FCNN** | $16 \pm 5$ | $2 \pm 1$ | $109 \pm 35$ | $12 \pm 2$ | $2 \pm 0$ |
| | **FCNN-GNN** | $39 \pm 1$ | $21 \pm 6$ | $48 \pm 2$ | $16 \pm 1$ | $14 \pm 3$ |
| Ours | **Res-GNN** | $36 \pm 7$ | $7 \pm 3$ | $41 \pm 7$ | $31 \pm 9$ | $7 \pm 2$ |
| | **BGNN** | $20 \pm 4$ | $2 \pm 0$ | $16 \pm 0$ | $21 \pm 7$ | $5 \pm 1$ |

The reason for improved efficiency is that **BGNN** and **Res-GNN** converge with a much fewer number of iterations as demonstrated in Figure 3. We plot RMSE on the test set during training for all models (with winning hyperparameters). We can see that **BGNN** converges within the first ten iterations (for $k = 20$), leading to fast training. In contrast, **Res-GNN** is similar in terms of convergence to **GNN** for the first 100 epochs, but then it continues decreasing RMSE unlike **GNN** that requires much more epochs to converge. This behavior is similar for other datasets (see Appendix F).

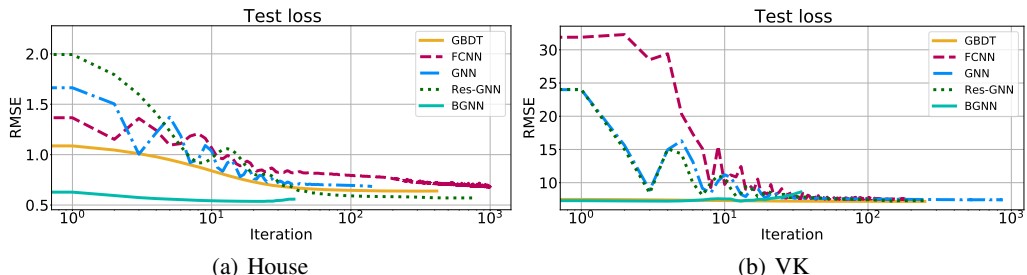

(a) House          (b) VK

Figure 3: RMSE on the test set during training for two node regression datasets.

## 4.5 VISUALIZING PREDICTIONS

To investigate the performance of **BGNN**, we plot the final predictions of trained models for observations in the training set. Our motivation is to scrutinize which points are correctly classified by different models. Figure 4 displays the predictions of **GBDT**, **GNN**, **Res-GNN**, and **BGNN** models as well as the true target value. To better understand the predictions of the **BGNN** model, in Figure 4(e) we show the values predicted by **GBDT** that was trained as a part of **BGNN**. This experiment is performed on **House** dataset, the plots for other datasets show similar trends and can be found in the supplementary materials.

Several observations can be drawn from these figures. First, the true target values change quite smoothly within local neighborhoods; however, there are a few outliers: single red points among many blue points and conversely. These points can mislead the model during the training, predicting

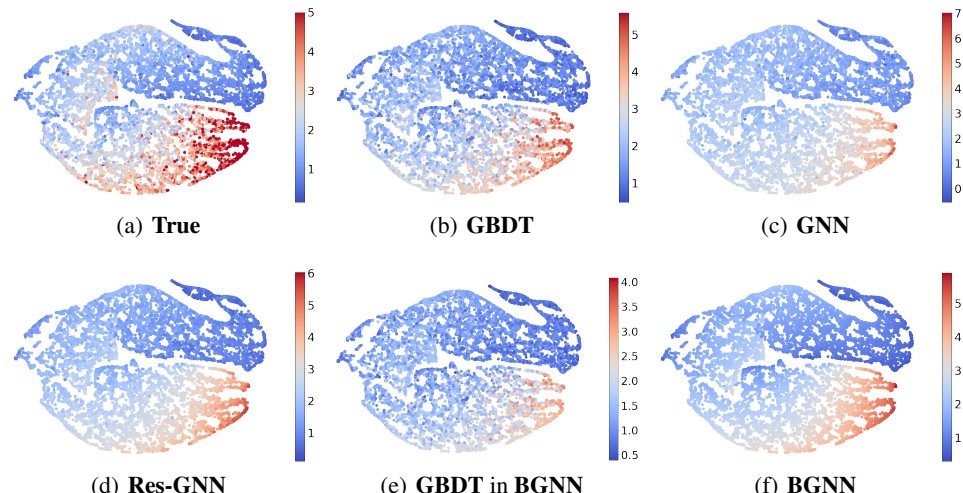

(a) **True**        (b) **GBDT**        (c) **GNN**

(d) **Res-GNN**        (e) **GBDT** in **BGNN**        (f) **BGNN**

Figure 4: **House** dataset. True labels and predictions by trained **GBDT**, **GNN**, **Res-GNN**, and **BGNN** models (training points only). Point coordinates correspond to **BGNN** learned representations in the first hidden layer. Color represents the final predictions made by each model.

the wrong target value for many observations in the outliers' local neighborhoods. Hence, it is important for a model to make smoothed predictions in the local neighborhoods.

Second, comparing the prediction spaces of **GBDT**, **GNN**, **Res-GNN**, and **BGNN** models we can observe that predictions for **GBDT** are much more grainy with large variations in the neighborhoods of the vertices (high quality images can be found in the supplementary materials). Intuitively, because the **GBDT** model does not have access to the graph structure, it cannot propagate its predictions in the nodes' vicinity. Alternatively, **GNN**, **Res-GNN**, and **BGNN** can extrapolate the outputs among local neighbors, smoothing out the final predictions as seen in Figures 4(c), 4(d), 4(f).

Third, focusing on the values of predictions (color bars on the right of each plot) of **GBDT**, **GNN**, and **BGNN** models we notice that the scale of final predictions for **GBDT** and **BGNN** models is closely aligned with the true predictions, while **GNN**'s predictions mismatch the true values by large margin. Our intuition is that the expressive power of **GBDT** to learn piecewise decision boundaries common in tabular datasets helps **GBDT** and **BGNN** to properly tune its final predictions with respect to the true range of values. In contrast, **GNN** relies solely on neural layers to learn complex decision rules.

Another observation comes from looking at the values predicted by **GBDT** trained as a part of **BGNN** (see Figure 4(e)). While this **GBDT** model is initialized using the true target labels, it was not forced to predict the target during the training. Interestingly, this model shows the same trend and clearly captures the regions on high/low target values. On the other hand, **GBDT** trained as a part of **BGNN** is much more conservative: on all datasets, the range of predicted values is significantly smaller than the true one. We hypothesize that **GBDT** is trained to scale its predictions to make them more suitable for further improvements by **GNN**.

## 5 CONCLUSION

We have presented **BGNN**, a novel architecture for learning on graphs with heterogeneous tabular node features. **BGNN** takes advantages of the GBDT model to build hyperplane decision boundaries that are common for heterogeneous data, and then utilizes GNN to refine the predictions using relational information. Our approach is end-to-end and can be incorporated with any message-passing neural network and gradient boosting method. Extensive experiments demonstrate that the proposed architecture is superior to strong existing competitors in terms of accuracy of predictions and training time. A possible direction for future research is to analyze whether this approach is profitable for graph-level predictions such as graph classification or subgraph detection.

ACKNOWLEDGMENTS

The authors thank the Anonymous Reviewers for their reviews and Anton Tsitsulin for kindly sharing VK data. Liudmila Prokhorenkova also acknowledge the financial support from the Ministry of Education and Science of the Russian Federation in the framework of MegaGrant 075-15-2019-1926 and from the Russian President grant supporting leading scientific schools of the Russian Federation NSh-2540.2020.1.

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

## A FURTHER RELATED WORK

To the best of our knowledge, there are no approaches combining the benefits of GBDT and GNN models for representation learning on graphs with tabular data. However, there are many attempts to adapt non-graph neural networks for tabular data or to combine them with gradient boosting in different ways.

Several works (Popov et al., 2019; Yang et al., 2018; Zhou & Feng, 2019; Feng et al., 2018; Hazimeh et al., 2020) attempt to mitigate the non-differentiable nature of decision trees. For example, Popov et al. (2019) proposed to replace hard choices for tree splitting features and splitting thresholds with their continuous counterparts, using $\alpha$-entmax transformation (Peters et al., 2019). While such an approach becomes suitable for a union of decision trees with GNN, the computational burden of training both end-to-end becomes a bottleneck for large graphs.

Another method (Badirli et al., 2020) uses neural networks as weak learners for the GBDT model. For graph representation problems such as node regression, one can replace standard neural networks with graph neural networks. However, training different GNN as weak classifiers at once would be exhaustive. Additionally, such a combination lacks some advantages of GBDT, like handling heterogeneous and categorical features and missing values. An approach called AdaGCN (Sun et al., 2019) incorporates AdaBoost ideas into the design of GNNs in order to construct deep models. Again, this method does not exploit the advantages of GBDT methods.

Finally, Li et al. (2019) investigated different ways of combining decision-tree-based models and neural networks. While the motivation is similar to ours — get the benefits of both types of models — the paper focuses specifically on learning-to-rank problems. Additionally, while some of their methods are similar in spirit to **Res-GNN**, they do not update GBDT in an end-to-end manner, which is a substantial contribution of the current research.

## B HYPERPARAMETERS

Parameters in brackets {} are selected by hyperparameter search on the validation set.

**LightGBM**: number of leaves is $\{15, 63\}$, $||\lambda||_2 = 0$, boosting type is gbdt, number of epochs is 1000, early stopping rounds is 100.

**CatBoost**: depth is $\{4, 6\}$, $||\lambda||_2 = 0$, number of epochs is 1000, early stopping rounds is 100.

**FCNN**: number of layers is $\{2, 3\}$, dropout is $\{0., 0.5\}$, hidden dimension is 64, number of epochs is 5000, early stopping rounds is 2000.

**GNN**: dropout rate is $\{0., 0.5\}$, hidden dimension is 64, number of epochs is 2000, early stopping rounds is 200. **GAT**, **GCN**, and **AGNN** models have two convolutional layers with dropout and ELU activation function (Clevert et al., 2016). **APPNP** has a two-layer fully-connected neural network with dropout and ELU activation followed by a convolutional layer with $k = 10$ and $\alpha = 0.1$. We use eight heads with eight hidden neurons for **GAT** model.

**Res-GNN**: dropout rate is $\{0., 0.5\}$, hidden dimension is 64, number of epochs is 1000, early stopping rounds is 100. We also tune whether to use solely predictions of **CatBoost** model or append them to the input features. **CatBoost** model is trained for 1000 epochs.

**BGNN**: dropout rate is $\{0., 0.5\}$, hidden dimension is 64, number of epochs is 200, early stopping rounds is 10, number of trees and backward passes per epoch is $\{10, 20\}$, depth of the tree is 6. We also tune whether to use solely predictions of **CatBoost** model or append them to the input features.

For all models, we also perform a hyperparameter search on learning rate in $\{0.1, 0.01\}$. Every hyperparameter setting is evaluated three times and an average is taken. We use five random splits for train/validation/test with 0.6/0.2/0.2 ratio. The average across five seeds is reported in the tables.

## C REGRESSION DATASETS

In **House** dataset (Pace & Barry, 1997), nodes are the properties, edges connect the proximal nodes, and the target is the property's price. We use the publicly available dataset (Pace & Barry, 1997) of

all the block groups in California collected from the 1990 Census. We connect each block with at most five of its nearest neighbors if they lie within a ball of a certain radius, as measured by latitude and longitude. We keep the following node features: *MedInc, HouseAge, AveRooms, AveBedrms, Population, AveOccup*.

**County** dataset (Jia & Benson, 2020) is a county-level election map network. Each node is a county, and two nodes are connected if they share a border. We consider node features coming from the 2016 year. These features include *DEM, GOP, MedianIncome, MigraRate, BirthRate, DeathRate, BachelorRate, UnemploymentRate*. We follow the setup of the original paper and select UnemploymentRate as the target label. We filter out all nodes in the original data if they do not have features.

**VK** dataset (Tsitsulin et al., 2018) comes from a popular social network where people are mutually connected based on friendships, and the regression problem is to predict the age of a person. We use an open-access subsample of the VK social network of the first 1M users.[3] Then, the dataset has been preprocessed to keep only the users who opt in to share their demographic information and preferences: *country, city, has_mobile, last_seen_platform, political, religion_id, alcohol, smoking, relation, sex, university*.

**Wiki** dataset (Rozemberczki et al., 2019) represents a page-page network on a specific topic (squirrels) with the task of predicting average monthly traffic. The features are bag-of-words for informative nouns (3148 in total) that appeared in the main text of the Wikipedia article. The target is the average monthly traffic between October 2017 and November 2018 for each article.

**Avazu** dataset (Song et al., 2019) represents a device-device network, with two devices being connected if they appear on the same site within the same application. For this dataset, the goal is to predict click-through-rate (CTR) for each device. We take the first 10M rows from the publicly available train log of user clicks.[4] We compute CTR for each device id and filter those ids that do not have at least 10 ad displays. We connect two devices if they had ad displays on the same site id from the same application id. The node features are anonymized categories: *C1, C14, C15, C16, C17, C18, C19, C20, C21*.

## D    CLASSIFICATION DATASETS

For node classification, we consider three types of node features: heterogeneous (**VK** and **House**), sparse (**Slap** and **DBLP**), and homogeneous (**OGB-ArXiv**).

For **House** and **VK**, we transform the original numerical target value with respect to the bin it falls to. More specifically, for **VK** we consider the classes $< 20, 20 - 25, 25 - 30, \ldots, 45 - 50, > 50$ for the age attribute. Similarly, for **House** dataset we replace the target value with the bin it falls to in the range $[1, 1.5, 2, 2.5]$. Hence, there are 7 and 5 classes for **VK** and **House**, respectively.

For datasets with sparse features, we consider two datasets coming from heterogeneous information networks (HIN), where nodes have a few different types. A common way to represent HIN is through meta-paths, i.e., a collection of all possible paths between nodes of a particular type. For example, for a citation network, one may specify paths of the type paper-author-paper (PAP) and the type paper-subject-paper (PSP). Then the original graph is approximated as several adjacency matrices for different types.

Table 5: Summary of classification datasets.

|  | SLAP | DBLP | OGB-ArXiv |
|---|---|---|---|
| **# Nodes** | 20419 | 14475 | 169343 |
| **# Edges** | 172248 | 40269 | 1166243 |
| **# Features** | 2701 | 5002 | 128 |
| **Classes** | 15 | 4 | 40 |
| **Min Class** | 103 | 745 | 29 |
| **Max Class** | 534 | 1197 | 27321 |

**DBLP** dataset (Ren et al., 2019) is a network with three node types (authors, papers, conferences) and four target classes of the authors (database, data mining, information retrieval, and machine learning). To obtain a single graph, we use the adjacency matrix for the relation APA, which closely reflects the relationships between authors. Each author has a bag-of-words representation (300 words) of all the abstracts published by the author. Furthermore, for every node, we compute the degrees for all

---

[3] https://github.com/xgfs/vk-userinfo
[4] https://www.kaggle.com/c/avazu-ctr-prediction

types of relations and append them as additional node features. Namely, we have two additional node features corresponding to degrees for paper nodes in APA and APCPA adjacency matrices.

**SLAP** dataset (Xiao et al., 2019) is a multiple-hub network in bioinformatics that contains node types such as chemical compound, gene, disease, pathway, etc. The goal is to predict one of 15 gene types. To obtain a single graph, we use the adjacency matrix for the relation GG between genes. Each gene has 3000 features that correspond to the extracted gene ontology terms (GO terms). As for **DBLP**, we compute the degrees for all types of relations and append them as additional node features.

As a dataset with homogeneous node features we consider **OGB-ArXiv** (Hu et al., 2020a). The node features correspond to a 128-dimensional feature vector obtained by averaging the embeddings of words in the title and abstract. Note that for this particular dataset we used the implementation of GAT[5] as a backbone architecture for **GNN**, **Res-GNN**, and **BGNN** models. This model scored the top place on the leaderboard.[6] A summary of statistics for all datasets is outlined in Table 5.

## E    COMPARISON OF GNN MODELS

In this section, we show the exact RMSE values and time for all tested GNN models on all regression datasets. We consider several state-of-the-art GNN models that include **GAT** (Veličković et al., 2018), **GCN** (Kipf & Welling, 2017), **AGNN** (Thekumparampil et al., 2018), and **APPNP** (Klicpera et al., 2019).

Table 6 demonstrates that for all considered models **BGNN** and **Res-GNN** achieve significant increase in performance compared to vanilla **GNN**. Additionally, end-to-end training of **BGNN** achieves typically better results than a straightforward implementation of **Res-GNN**.

Table 6: Summary of our results for different GNN architectures for node regression.

| | Method | House RMSE | House Time (s) | County RMSE | County Time (s) | VK RMSE | VK Time (s) | Wiki RMSE | Wiki Time (s) | Avazu RMSE | Avazu Time (s) |
|---|---|---|---|---|---|---|---|---|---|---|---|
| **GAT** | **GNN** | $0.54 \pm 0.01$ | $35 \pm 2$ | $1.45 \pm 0.06$ | $19 \pm 6$ | $7.22 \pm 0.19$ | $42 \pm 4$ | $45916 \pm 4527$ | $15 \pm 1$ | $0.113 \pm 0.01$ | $9 \pm 2$ |
| | **Res-GNN** | $0.51 \pm 0.01$ | $36 \pm 7$ | $1.33 \pm 0.08$ | $7 \pm 3$ | $7.07 \pm 0.20$ | $41 \pm 7$ | $46747 \pm 4639$ | $31 \pm 9$ | $0.109 \pm 0.01$ | $7 \pm 2$ |
| | **BGNN** | $0.5 \pm 0.01$ | $20 \pm 4$ | $1.26 \pm 0.08$ | $2 \pm 0$ | $6.95 \pm 0.21$ | $16 \pm 0$ | $49222 \pm 3743$ | $21 \pm 7$ | $0.109 \pm 0.01$ | $5 \pm 1$ |
| **GCN** | **GNN** | $0.63 \pm 0.01$ | $28 \pm 0$ | $1.48 \pm 0.08$ | $18 \pm 7$ | $7.25 \pm 0.19$ | $38 \pm 0$ | $44936 \pm 4083$ | $13 \pm 3$ | $0.114 \pm 0.02$ | $12 \pm 6$ |
| | **Res-GNN** | $0.59 \pm 0.01$ | $25 \pm 2$ | $1.35 \pm 0.09$ | $11 \pm 5$ | $7.03 \pm 0.20$ | $52 \pm 6$ | $44876 \pm 3777$ | $21 \pm 5$ | $0.111 \pm 0.02$ | $9 \pm 6$ |
| | **BGNN** | $0.54 \pm 0.01$ | $41 \pm 15$ | $1.33 \pm 0.13$ | $12 \pm 8$ | $7.12 \pm 0.21$ | $76 \pm 6$ | $47426 \pm 4112$ | $22 \pm 11$ | $0.107 \pm 0.01$ | $4 \pm 1$ |
| **AGNN** | **GNN** | $0.59 \pm 0.01$ | $38 \pm 5$ | $1.45 \pm 0.08$ | $28 \pm 3$ | $7.26 \pm 0.20$ | $48 \pm 3$ | $45982 \pm 3058$ | $19 \pm 5$ | $0.113 \pm 0.02$ | $14 \pm 8$ |
| | **Res-GNN** | $0.52 \pm 0.01$ | $33 \pm 4$ | $1.3 \pm 0.07$ | $16 \pm 4$ | $7.08 \pm 0.20$ | $51 \pm 15$ | $46010 \pm 2355$ | $24 \pm 3$ | $0.111 \pm 0.02$ | $7 \pm 2$ |
| | **BGNN** | $0.49 \pm 0.01$ | $34 \pm 4$ | $1.28 \pm 0.08$ | $3 \pm 1$ | $6.89 \pm 0.21$ | $25 \pm 4$ | $53080 \pm 5117$ | $47 \pm 37$ | $0.108 \pm 0.02$ | $5 \pm 1$ |
| **APPNP** | **GNN** | $0.69 \pm 0.01$ | $68 \pm 1$ | $1.5 \pm 0.11$ | $34 \pm 10$ | $13.23 \pm 0.12$ | $81 \pm 3$ | $53426 \pm 4159$ | $49 \pm 26$ | $0.113 \pm 0.01$ | $24 \pm 15$ |
| | **Res-GNN** | $0.67 \pm 0.01$ | $58 \pm 12$ | $1.41 \pm 0.12$ | $19 \pm 10$ | $13.06 \pm 0.17$ | $76 \pm 11$ | $53206 \pm 4593$ | $66 \pm 27$ | $0.110 \pm 0.01$ | $15 \pm 10$ |
| | **BGNN** | $0.59 \pm 0.01$ | $21 \pm 7$ | $1.33 \pm 0.10$ | $17 \pm 6$ | $12.36 \pm 0.14$ | $50 \pm 6$ | $54359 \pm 4734$ | $30 \pm 13$ | $0.108 \pm 0.01$ | $6 \pm 1$ |

---

[5]https://github.com/Espylapiza/dgl/blob/master/examples/pytorch/ogb/ogbn-arxiv/models.py

[6]https://ogb.stanford.edu/docs/leader_nodeprop/#ogbn-arxiv

## F    LOSS CONVERGENCE

In Figure 5, we plot RMSE on the test set during training for the remaining datasets — **County**, **Wiki**, and **Avazu**. These results confirm that **BGNN** converges to its optimal value within the first ten iterations (for $k = 20$). Note that on the **Wiki** dataset, similarly to Figure 3, **Res-GNN** convergence is similar to **GNN** for the first 100 iterations and then the loss of **Res-GNN** decreases faster than of **GNN**.

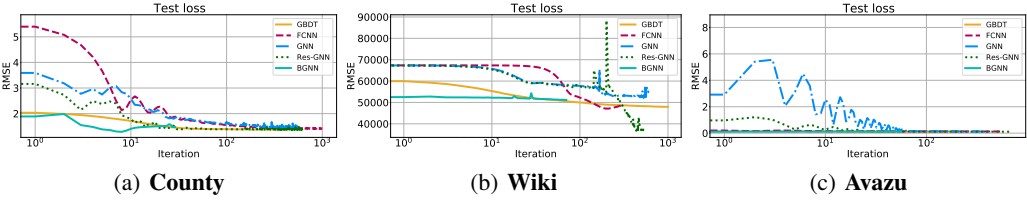

(a) **County**           (b) **Wiki**           (c) **Avazu**

Figure 5: Summary of RMSE of test set during training for node regression datasets.

