# OpenReview forum: "Boost then Convolve: Gradient Boosting Meets Graph Neural Networks"
_ICLR.cc/2021/Conference — ICLR 2021 Poster_

### Official Review · AnonReviewer3 · 2020-10-16
**A novel gradient boosted decision tree model that works better on graph representation data**

**Rating:** 5
**Confidence:** 4

**Review:**

Review: This paper proposes a fusion of GBDT and graph neural network that works on graphs with heterogeneous tabular features. Previous approaches are computationally heavy and do not consider graph-structured data and suffer from lack of relational bias imposed in GNNs. The proposed method is a new ensemble tree method which alternates between functional gradient step in GBDT (which train on the current latent features) and SGD training of graph neural network (to generate the latent features which are fed into the subsequent trees).

Reasons for score:

Overall, I am leaning toward rejecting this paper. Overall, the paper is an empirically motivated paper focusing on the construction of pipelines which depend on already established methods, GBDT and GNN. Having advantages of both methods is nice, but it is hard to glean why this proposed method works well on theoretical basis.

+Positives:
- The proposed end-to-end training combining GBDT and GNN is easy to implement and clearly described.

- improving GBDT to work on graph-structured data is a novel idea.

- Comparison in the experimental section considers a handful of state-of-the-art approaches.

Concerns:
- Experimental results are mainly concerned with regression tasks. Having some results on classification would be nice.

- The contribution of the paper is mainly empirical and offers little intuition on why the proposed method improves upon GNN and GBDT.

---

> ### Author Response · Authors · 2020-11-21
> **Response to Reviewer 3**
>
> Thank you very much for your feedback and comments. We updated the paper to address the points you raised.
>
> > Experimental results are mainly concerned with regression tasks. Having some results on classification would be nice.
>
> We provide a separate extensive evaluation of our approach for node classification tasks in Section 4.2. Part of these results was available in the original text (Slap and DBLP datasets), but due to the limit of space, we placed it in Appendix. We currently placed this analysis in the experimental section of the main text and also made an extensive evaluation with hyperparameter tuning on three other datasets: House, VK, and OGB-ArXiv.
>
> We show that Res-GNN and BGNN approaches continue to work well on classification datasets with heterogeneous features. For datasets Slap and DBLP that have sparse homogeneous features, we show that BGNN achieves the top-2 performance with a significant margin over the pure GNN model. On the OGB-ArXiv dataset, the node features are pre-trained word2vec embeddings and hence are homogeneous. As described in the introduction, the GBDT approach typically works better in heterogeneous settings. Therefore, we do not expect that adding a GBDT component to GNN can lead to better results than pure GNN. Rather predictions of GBDT as input features for GNN would be redundant. That’s indeed the case, and the GNN model achieves the top performance.
>
> > The contribution of the paper is mainly empirical and offers little intuition on why the proposed method improves upon GNN and GBDT.
>
> To address this concern we investigated the predictions made by different models with respect to their positions in the graph. The results can be found in Section 4.5 and supplementary materials, which contain high-quality images for different datasets (which we exclude from the main paper due to the size of the images).
> There are several observations that can be made by these figures that shed the light on the performance of different models. First, the true target labels are distributed smoothly in the graph (i.e. local neighbors tend to have similar target values); however, there are several outliers with significantly different target values than of their neighbors. Such outliers can mislead models to predict the wrong target value for several nodes in the vicinity of the outliers and therefore it is important to make smooth predictions across all points. We also note that such outliers are the artifact of many real datasets and hence it is important for the model to be resistant to them.
>
> Our second observation is that in comparison between GBDT, GNN, Res-GNN, and BGNN models, GBDT approach has the most uneven distribution of predictions that could be explained by the fact that it does not utilize the topology of the network (i.e. adjacent nodes can have significantly different predictions). On the other hand, GNN predictions are usually smooth (due to usage of the graph signal), but the learned decision boundaries are weaker than of those learned by GBDT, which can be seen by looking at the values of the predictions. As outlined in the paper, GBDT’s core strength is on processing complex hyperplane decision boundaries, and therefore BGNN is a good trade-off between utilizing the expressive power of GBDT and learning the graph structure of GNN. Additionally, it is important to note that BGNN is trained end-to-end, which better learn the interaction between the graph structure and tabular node features than Res-GNN. We provide additional insights into the performance in Section 4.5.
>
> All our experiments confirm the intuition in which cases the proposed approach is supposed to work: BGNN is especially successful on graph datasets with heterogeneous features. This approach outperforms pure GBDT since GBDT cannot account for the graph structure, and in BGNN the predictions of GBDT are further improved by GNN. Likewise, this approach outperforms GNN since GBDT is known to be more suitable for tabular heterogeneous features.

---

> > ### Comment · AnonReviewer3 · 2020-11-22
> > **Some comments on accuracy vs efficiency tradeoff**
> >
> > Thank you for your clarifications on experimental results and contributions of this paper. I have some additional questions:
> >
> > 1. On classification results (Table 3), the newly proposed method does not perform the best for some datasets. I would think there needs to be some tradeoff analysis done to argue why for some of the datasets loss in accuracy is okay.
> >
> > 2. In addition, it would be great if the authors could clarify what is meant by "running time" in Section 4.4. Is this pertaining to the prediction time or also includes the training time? How does the size of the model different for all the methods tested in this paper and does it explain the tradeoff between accuracy and efficiency?

---

> > > ### Author Response · Authors · 2020-11-23
> > > **Response to the comments**
> > >
> > > Thank you for the feedback and additional comments, which we discuss below.
> > >
> > > > On classification results (Table 3), the newly proposed method does not perform the best for some datasets.
> > >
> > > To elaborate on the results in Table 3, the principal difference for the results in node classification is the diversity of the node features in the datasets. In particular, House and VK datasets have heterogeneous features, i.e. features of different scale, type, and meaning. For example, the node features for VK dataset include country, followers_count, last_seen_time, family_status, gender, and 9 other features (described in Appendix C). On the other hand, Slap and DBLP have sparse bag-of-words features (keywords that appear in the ontologies or abstracts). For example, each node in DBLP has 5002 features; however, only around 10-30 of these features are non-zeros. Furthermore, OGB-ArXiv dataset has homogeneous node embeddings (average word representations in the title and abstract). While representation in OGB-ArXiv is dense, all features are trained jointly and have the same scale, type, and meaning.
> > >
> > > As our approach includes GBDT that processes the original features, it is expected to work well when the classification task is more amenable for GBDT rather than fully-connected neural network (FCNN) model. How to understand that a given classification dataset is more suitable for GBDT? In general, it is an open problem; however, there are several indications that GBDT works better on tabular datasets with heterogeneous features [1,2,3]. In our experiments (Table 3), we can envisage if it is worth using our BGNN model by comparing performance of GBDT, FCNN, and GNN models.
> > >
> > > * On heterogeneous datasets House and VK, we have GBDT > FCNN (i.e., datasets are suitable for GBDT) and GNN > FCNN (i.e., graph structure helps). Hence, BGNN outperforms other approaches, as expected.
> > > * On sparse datasets Slap and DBLP, we have GNN < FCNN. This means that graph structure does not provide additional information to FCNN and hence it cannot improve GBDT too.
> > > * On homogeneous dataset OGB-ArXiv, we have GBDT < FCNN. This means that node features are more suitable for neural networks (as they originate from a pretrained neural network). Therefore, GBDT part of BGNN model is not expected to work better than GNN, as confirmed in Table 3.
> > >
> > > Our conclusion is that BGNN is a good model for graph datasets with heterogeneous tabular features that appear frequently in real applications (e.g., social networks, recommender systems, credit scoring), but have not been studied in the academic literature in great detail.
> > >
> > > > In addition, it would be great if the authors could clarify what is meant by "running time" in Section 4.4. Is this pertaining to the prediction time or also includes the training time? How does the size of the model different for all the methods tested in this paper and does it explain the tradeoff between accuracy and efficiency?
> > >
> > > Thank you for this comment and we corrected our terminology of "running time" in the paper. In section 4.4, by running time we mean the overall training time for a model to converge, i.e. the clock time for all epochs during the training, considering early stopping. It may seem that BGNN can introduce significant overhead in training time since each iteration of BGNN training includes both GNN backpropagation steps and training several decision trees. However, we found that BGNN converges much faster than GNN model: usually, BGNN requires 5-10 epochs to converge, while GNN requires thousands of epochs to converge. We believe that the reason for faster convergence is that GBDT predictions used for the initialization of node features in BGNN lie within a much closer region to the optimal values than the original node features.
> > >
> > > In contrast to training time, prediction time is much smaller (less than a second) and is similar across all models.
> > >
> > > We also note that GNN part has the same architecture with the same number of parameters for all models (GNN, Res-GNN, BGNN), and GBDTs are very fast to compute compared to GNN. Therefore, Section 4.4 shows that in general, it is faster to train BGNN model than GNN, despite the fact that BGNN has an additional GBDT part.
> > >
> > > [1] Neural Oblivious Decision Ensembles for Deep Learning on Tabular Data, ICLR 2020
> > >
> > > [2] Lessons from 2 Million Machine Learning Models on Kaggle https://www.kdnuggets.com/2015/12/harasymiv-lessons-kaggle-machine-learning.html
> > >
> > > [3] Historical data science trends on Kaggle. https://www.kaggle.com/shivamb/data-science-trends-on-kaggle

---

### Official Review · AnonReviewer1 · 2020-10-29
**A very creative method for training combinations GNNs and Gradient Boosted Decision Trees end-to-end with strong results!**

**Rating:** 9
**Confidence:** 4

**Review:**

This paper presents a new architecture called BGNN (Boost-GNN), which combines the benefits of GNNs (Graph Neural Nets) with GBDTs (Gradient Boosted Decision Trees).

Basic idea:
* GBDTs work well with *heterogeneous* tabular data.
* GNNs work well on *graphs* with *homogeneous* sparse features.
* BGNNs work well on *graphs* where the nodes contain *heterogeneous* tabular data.
* BGNNs is optimized end-to-end and seems to obtain great SOTA results!

An example of the data BGNN works well on is in social networks. E.g. each node could be a person with heterogeneous characteristics such as age, gender, graduation date.

The main trick is in how to train this end-to-end effectively. This is done by iteratively adding trees that fit the GNN gradient updates, allowing the GNN to *backpropagate* into the GBDT.

More in detail:
* GBDTs make a prediction for each node in the graph.
* The GNN reads the output of the GBDTs and corrects the predictions based on the graph structure.
* When doing gradient descent on the GNN the authors critically also optimize with respect to the input features. The difference between the optimized input features and the original input features becomes the new objective for next round of update on the decision trees.

Results seem very strong across all tasks considered in the paper.

Strengths:
* The paper presents an interesting and convincing analysis of why GBDTs are so advantageous for tabular data, while GNNs are the best choice for graph data.
* The combination of GNNs and GBDTs is not trivial. Several papers are cited attempting the combination of neural nets and gradient boosting, but they are reported to be computationally more expensive and not as powerful.
* The detail about computing updated targets for GBDTs while doing gradient descent on the GNN seems particularly interesting and creative.
* The results obtained by this paper are impressive, surpassing SOTA models by considerable margins.
* The paper also finds that representations learned with BGNN have more discernible structure, suggesting that they are more interpretable.

---

> ### Author Response · Authors · 2020-11-21
> **Response to Reviewer 1**
>
> Thank you for your valuable review and accurate interpretation of the main ideas of our paper. In addition, we provide experiments on five node classification datasets with various properties (Section 4.2) and interpretation of the predictions made by all models on these datasets (Section 4.5), which further provides insights into the inner workings of the proposed models.

---

### Official Review · AnonReviewer4 · 2020-10-30
**A simple method combining GBDT and GNNs to deal with graph datasets with tabular heterogeneous features**

**Rating:** 6
**Confidence:** 3

**Review:**

This paper aims to learn from graphs with tabular node features. Existing methods are only designed to handle either tabular data, such as gradient boosting decision tree (GBDT), or graph-structured data, such as graph neural networks (GNNs). This paper naturally extends GBDT to deal with graph-structured data and train it together with GNN in end-to-end fashion.

This paper is clearly written and easy to follow. I enjoyed reading this paper. The idea is clear and well-motivated. The way of combining GBDT and GNN looks natural and reasonable. The essence of GBDT is to approximate the gradient of the undifferentiable part of the model by selecting a weak learner h from H (eq. 2). This proposed training method seems valid and effective on semi-supervised regression.

My main concern is the proposed method is not working well on semi-supervised classification tasks. It performs worse than CatBoost (pure GBDT method without using the graph) on the two datasets Slap and DBLP for semi-supervised node classification, which is probably why the authors put the experiments in the Appendix.

Since node classification is a benchmark task and, in my opinion, more common and important than semi-supervised regression, I have some doubts of the practicality of the proposed method.

============Post Rebuttal=============================

Thanks for the additional experiments and the updates. The new results are informative.

---

> ### Author Response · Authors · 2020-11-21
> **Response to Reviewer 4**
>
> Thank you for your valuable review and we appreciate your specific suggestions for improving the paper.
>
> In particular, we include Section 4.2 with node classification on five datasets with various properties. In addition to Slap and DBLP datasets (previously placed in the Appendix), we now added three more datasets, VK, House, and OGB-ArXiv. We note that VK and House have tabular node features, while OGB-ArXiv has homogeneous features.
>
> BGNN is expected to outperform other models on graph datasets containing tabular heterogeneous features since GBDTs are especially successful on such data. Agreeing with our expectations, BGNN outperforms the baselines by a large margin, confirming our intuition that BGNN is a good algorithm for graph datasets with tabular heterogeneous features.
>
> We note that Slap and DBLP datasets are particularly challenging for GNN: it turns out that GNN is outperformed by a simple FCNN model that does not use any graph structure. Hence, adding GNN to GBDT does not improve GBDT. However, our model is still close to the best one in this case.
>
> Also, we performed an additional experiment on the ArXiv dataset from Open Graph Benchmark. We note that OGB datasets have homogeneous features such as pre-trained word2vec embeddings. For homogeneous data, it is known that neural approaches usually work better than GBDT, as we can observe for the OGB-ArXiv dataset (FCNN > GBDT). As a result, Res-GNN and BGNN models (which use GBDT as a part of the model) achieve slightly worse performance than GNN but better than GBDT or MLP.
>
> Overall, on both node regression and classification problems, our proposed approach consistently outperforms other strong baselines on datasets with heterogeneous features. Such datasets have been historically under the radar in the graph research community, although they are popular in real applications (e.g. social networks, recommender systems, credit scoring) and we hope our work can be of high practical importance for such use cases.

---

### Official Review · AnonReviewer2 · 2020-10-31
**Review for "Boost then Convolve: Gradient Boosting Meets Graph Neural Networks"**

**Rating:** 7
**Confidence:** 4

**Review:**

**Summary**
The paper proposes a GNN model by incorporating gradient boosting. In the proposed BGNN, the input feature on the graph is learned by the gradient boosting model. The processed feature then becomes a new feature for a GNN model following the gradient boosting. Several experiments demonstrate the improvement of the performance for a tabular feature and graph-structured datasets. The running time of Res-GNN/BGNN is shown to have a significant reduction as compared to the plain GNN methods.

**Comments and questions**
1.  The paper proposed a simple model combining Boosting and GNN methods, which can effectively learn the heterogeneous features of graph-structured data.
2.  The performance of BGNN, including the training speed and regression error, is good as compared to the GNNs for datasets with tabular features.
3.  What is the performance of the proposed model on other node property prediction tasks such as Open Graph Benchmark?
4. The BGNN has two stages: a gradient boosting for graph features and graph neural networks for graph-structured data, including features plus graph structure. So, the computational time should be the sum of time of GBDT and GNN. Why is the time becoming smaller than the plain GNN for the same datasets?
5. How many layers do the BGNN exactly use for experiments?
6. Will the GBDT apply to link prediction tasks?

---

> ### Author Response · Authors · 2020-11-21
> **Response to Reviewer 2**
>
> Thank you for your valuable comments and specific questions, which helped us to clarify our contributions. Please, see below the answers to the questions as well as the requested additional experiments on property prediction tasks.
>
> Q. What is the performance of the proposed model on other node property prediction tasks such as Open Graph Benchmark?
>
> A. In addition to the previous experiments on node classification (that were described in the appendix), we provided additional experiments for node property predictions, included in Open Graph Benchmark. You can find a thorough subsection on node classification in Section 4.2. We make a hyperparameter tuning and extensive evaluations similar to previous experiments in the paper.
>
> We consider OGB-ArXiv dataset as the other datasets are bigger and thus require more time to do a fair comparison of the models. We note that OGB-ArXiv has homogeneous features such as pre-trained word2vec embeddings. For homogeneous data, it is known that neural approaches usually work better than GBDT, as we can observe for the OGB-ArXiv dataset (MLP > GBDT). As a result, Res-GNN and BGNN models (which use GBDT as a part of the model) achieve slightly worse performance than GNN but better than GBDT or MLP.
> On the other hand, performance on other datasets with heterogeneous features demonstrates that Res-GNN and BGNN can achieve higher performance than both GBDT/MLP (which do not use graph structure) and GNN (which does not use GBDT strengths).
>
>
>
> Q. The BGNN has two stages: a gradient boosting for graph features and graph neural networks for graph-structured data, including features plus graph structure. So, the computational time should be the sum of time of GBDT and GNN. Why is the time becoming smaller than the plain GNN for the same datasets?
>
> A. The running time for all models is the total time for training, considering early stopping. BGNN model attains lower running time because the number of epochs necessary to converge is much smaller (Figures 3 and 5), usually within the first ten epochs, while GNN and other models may require thousands of epochs to converge. We believe that the reason for faster convergence is that GBDT predictions used for the initialization of node features for GNN lie within a much closer region to the optimal values than the original node features.
>
>
>
> Q. How many layers do the BGNN exactly use for experiments?
>
> A. We used three hidden layers for all GNN models and the corresponding parts in BGNN. For the OGB-ArXiv dataset, we used the top-1 implementation in the leaderboard (https://github.com/Espylapiza/dgl/blob/master/examples/pytorch/ogb/ogbn-arxiv/models.py).
>
>
>
> Q. Will the GBDT apply to link prediction tasks?
>
> A. To the best of our knowledge, GBDT is not usually applied for link prediction tasks since it is hard for GBDT to account for the graph structure. However, as a part of BGNN, GBDT can be used for this purpose. Indeed, as in node prediction tasks, we could apply GBDT to the original features and then train link-predicting-GNN on GBDT outputs. The only technical difficulty here is how to initialize GBDT model (for node prediction, we train the first several trees using target labels in the training dataset). We have some preliminary thoughts on how to solve this issue, but this direction deserves more time, and we will keep it for future work.

---

### Author Response · Authors · 2020-11-24
**Summary of changes**

We would like to thank the reviewers for their valuable comments. We hope that we have satisfactorily addressed the concerns during this rebuttal process. If there are any more comments or questions, we will be happy to address them.

We made the following changes to the paper:
- While our paper was mostly focused on node regression (since there are several heterogeneous regression datasets publicly available), we conducted additional analysis on node classification tasks, including a dataset from Open Graph Benchmark. Our experiments confirm that BGNN is very successful in tasks with heterogeneous tabular data. Interestingly, on homogeneous data, BGNN is still very close to the best-performing algorithms.
- We clarified that our time analysis concerns training time.
- To better understand the differences between the models, we added Section 4.5 to visualize predictions and discuss their properties.

Sincerely, Authors

---

### Comment · ~Sergei_Ivanov2 · 2021-01-21
**Code and datasets**

We release the code and datasets here: https://github.com/nd7141/bgnn

---

### Decision · Program_Chairs · 2021-01-07
**Final Decision**

**Decision:**

Accept (Poster)

**Comment:**

The paper proposes an interesting architecture that dues Graph Neural Networks (GNN) and Gradient Boosting Decision Tree. This new architecture works on graphs with heterogeneous tabular features and BGNNs work well on graphs where the nodes contain heterogeneous tabular data and is optimized end-to-end and seems to obtain great SOTA results. End to end learning is done by iteratively adding trees that fit the GNN gradient updates, allowing the GNN to backpropagate into the GBDT. All reviewers agreed that the idea is interesting, the paper is well-written, and the results found in the paper are impressive. In addition, author response satisfactorily addressed most of the points raised by the reviewers, and most of them increased their original score. Therefore, I recommend acceptance.